# Growth Performance and Flesh Quality of Sea Bass (*Dicentrarchus labrax*) Fed with Diets Containing Olive Oil in Partial Replacement of Fish Oil—With or Without Supplementation with *Rosmarinus officinalis* L. Essential Oil

**DOI:** 10.3390/ani14223237

**Published:** 2024-11-12

**Authors:** Simona Tarricone, Marco Ragni, Claudia Carbonara, Francesco Giannico, Francesco Bozzo, Alessandro Petrontino, Anna Caputi Jambrenghi, Maria Antonietta Colonna

**Affiliations:** 1Department of Soil, Plant and Food Sciences, University of Bari Aldo Moro, Via Amendola 165/A, 70125 Bari, Italy; simona.tarricone@uniba.it (S.T.); marco.ragni@uniba.it (M.R.); francesco.bozzo@uniba.it (F.B.); alessandro.petrontino@uniba.it (A.P.); anna.caputijambrenghi@uniba.it (A.C.J.); mariaantonietta.colonna@uniba.it (M.A.C.); 2Department of Veterinary Medicine, University of Bari Aldo Moro, S.P. per Casamassima km 3, 70010 Valenzano, Italy

**Keywords:** *Dicentrarchus labrax*, rosemary oil, fish oil replacement, olive oil, flesh quality, sensory evaluation

## Abstract

The Mediterranean area produces the largest quantity of olive oil worldwide. Olive oil is used in feed formulations for several animal species, being economically more feasible and sustainable than other vegetable oils. Reducing the dependence on fish oil is critical for the sustainable development of the aquaculture industry; therefore, there is increased interest towards the inclusion of alternative vegetable proteins and oils in fish diets. Moreover, some essential oils that are rich in antioxidants and bioactive compounds, such as rosemary essential oil, can be supplemented in the fish diet to decrease the occurrence of oxidative processes and preserve the shelf life of fillet which contains high levels of polyunsaturated fatty acids. Any innovation in fish feeding requires a careful analysis of the expected benefits in terms of increased efficiency, cost reduction, and fish quality improvement.

## 1. Introduction

Fish meal and fish oil are important ingredients in the diet of carnivorous marine fish such as the sea bass due to their high digestibility and fatty acid composition; they are rich in n-3 long-chain polyunsaturated fatty acids, which are crucial for the growth and health of fish. Nowadays, the aquaculture industry uses approximately 40% and 60% of the global production of fish meal and fish oil, respectively [1,2]. In the next ten years, it has been estimated that fish oil production might not cover all the needs of the fish feed industry, which will lead to a sharp rise in the price of fish oil [3]. Reducing the dependence on fish oil is critical for the sustainable development of the aquaculture industry; therefore, there is increased interest towards the inclusion of alternative vegetable proteins and oils in fish diets [4,5].

Among vegetable oils that may be tested in alternative to fish oil, we have focused our attention on olive oil, since Apulia is the Italian region leader in olive oil production; therefore, in addition to its suitability in terms of chemical and fatty acid composition, this oil may be considered a sustainable alternative to fish oil also under an environmental and economic point of view. Olive oil has moderate levels of 18:2n-6, low levels of 18:3n-3 and high levels of 18:1n-9, which is considered a preferred substrate for energy production [6,7]. In European sea bass, a diet substituting up to 60% of the FO with OO did not affect fish growth [8]. However, higher levels of vegetable oil caused lower growth rates in European sea bass and Gilthead sea bream, presumably because the low FO proportion was insufficient to meet the essential fatty acids (EFA) requirement [8,9]. Furthermore, it has been reported that the replacement of more than 60% of FO with a single vegetable oil may affect immunity or stress resistance in gilthead sea bream. Thus, EFA requirements and the health of the fish should be considered when using plant oils in marine fish diets [9].

Being rich in fatty acids of the n-6 and n-3 series, the olive oil present in the fish feed may undergo oxidation; therefore, the addition of antioxidants to the feed may be useful in preventing the occurrence of oxidative processes. In this regard, the dietary administration of some plant-derived extracts rich in bioactive molecules is a recent strategy attracting growing interest [10,11], since these compounds show antioxidant properties and do not affect the growth or the morphometric parameters of fish. Some essential oils can be supplemented in the diet of sea bass to decrease the occurrence of oxidative processes and preserve the shelf-life of the fillet which contains high levels of polyunsaturated fatty acids [12]. Rosemary essential oil (RO) is usually extracted by an easy-to-handle steam distillation procedure, which provides an appreciable product on the basis of sensory evaluation. It is a natural food additive used for food preservation since it exerts antioxidant, antimicrobial, antiviral, antimycotic, and healing properties [13,14]. These antioxidant proprieties may be particularly useful in reducing the occurrence of fillet lipid oxidation following administration of diets rich in unsaturated fatty acids.

Furthermore, any innovation requires a careful analysis of the expected benefits to be integrated into the production process before replacing a conventional method. It is therefore appropriate to identify the main advantages that process innovation may bring, such as increased efficiency, cost reduction, and quality improvement. Efficiency analysis has become popular both as a research instrument and a practical decision-support tool. To define efficiency, one of the main used approaches is the structural one that starts from an economic optimization behavior and characterizes inefficiency in terms of deviation from this economic model [15,16,17].

The aim of this study was to investigate the effect of partial replacement of fish oil with olive oil, with or without the addition of rosemary essential oil as antioxidant, into the diet of farmed sea bass, on growth performance, fillet quality, shelf life, and sensory parameters. An economic analysis is provided in order to describe the economic effect on efficiency and management due to the replacement of fish oil with olive oil, and supplementation with rosemary essential oil in the sea bass diet.

## 2. Materials and Methods

### 2.1. Preparation of Feed and Dietary Regimen

Two isoenergetic and isoproteic feeds, with a lipid and protein content of 18.65 ± 0.23% and 48.13 ± 0.70%, respectively, were prepared for the trial. Each feed was added with 200 ppm of essential oil of *Rosmarinus officinalis* (Farmalabor S.R.L., Canosa di Puglia, BAT, Italy) whose chemical composition in constituents is shown in Table 1.

The ingredients were mixed, and the diets were prepared by cooking extrusion, using a semi-industrial extruder (E 19/25 D Brabender, Duisburg, Germany). The rosemary essential oil (RO) was added to the diets before the extrusion process. Since this product is fat-soluble, it was dissolved in fish oil before mixing it with the rest of the feed ingredients. The diet clumped together and formed a homogenous dough after the slow addition of water. Then, the dough was extruded by a mill machine (Fish Feed Pellet Making Machine, Richi machinery, Kaifeng, China) to obtain pellets with a diameter of 4 mm. For 24 h, the pellets were placed at room temperature to dry; then, they were broken and stored at −20 °C. Samples of all the diets were analyzed for chemical composition [18] and for the fatty acid profile, and the results are shown in Table 2.

### 2.2. Fish Sample Collection

The trial was carried out using 504 European Sea basses (*Dicentrarchus labrax*) reared in a commercial farm located in Apulia (South Italy; 41°37′14″ N, 15°56′57″ E). Fish with an average initial weight of 320 g were randomly distributed in 12 tanks of 2 m^3^/tank (42 fish/tank), with each diet tested in triplicate (three tanks per dietary treatment). The tanks were supplied with continuous seawater flow (34%; 10 L/min), with temperature ranging from 19 to 22 °C, oxygen from 4 to 6 ppm (sensors of Softmakers S.R.L., Cittadella, PD, Italy), and natural photoperiod conditions of approximately 12 h light/12 h darkness. During the feeding trial, sea bass were fed by hand to apparent visual satiation, once daily, 6 days per week. The daily feed consumption was recorded to calculate the feed conversion rate (FCR).

Fish were treated according to the “Council Directive 86/609 EEC for the protection of animals used for experimental and other scientific purposes” and to the “Ethical Justification for the Use and Treatment of Fishes in Research” [19].

The growth trial lasted 200 days; at the end of the experiment, all fish were individually weighed, and 24 fish per diet were slaughtered. The fish were chosen from the three tanks corresponding to each experimental group (eight subjects from each tank) and killed by means of hypothermia, using a mix of water and ice (1:3) according to the laws in force [20]. The fish were immediately placed in flaked ice and transported to the laboratory in refrigerated conditions. In order to reproduce the storage conditions performed in fish markets, where fish are stored and sold whole and ungutted, fish were refrigerated at 2 ± 1 °C for 0, 3, 10 and 17 days, placed in polystyrene boxes provided with outlets for water drainage and covered with flaked ice inside a plastic bag, as previously described by Álvarez et al. [12]. The ice/fish ratio (1:1) was kept constant throughout the storage period, by re-icing every day the fish boxes and checking the temperature in order to ensure proper fish storage.

The following parameters were calculated [21]:Condition Factor (K) = 100 × (Body Weight/Total Length^3^);
Edible yield = 100 × (Edible Part Weight/Body Weight);
Viscerosomatic Index (VSI) = [wet weight of viscera and associated fat/wet body weight] × 100;
Hepatosomatic index (HSI) = [wet weight of liver/wet body weight] × 100;
Average daily weight gain (AWG) = [W2 − W1]/T;
where W1 is the average body weight at the start of the trial, W2 is the average body weight at the end of trial, and T is the number of days of the trial.
Specific growth rate (SGR) = 100 × (ln W2 − ln W1)/T;
Survival (%) = [number of live fish harvested/number of fish stocked] × 100; 
Feed conversion rate (FCR) = Total Feed Intake/Weight Gain.

### 2.3. Analytical Determinations

At each time of storage (0, 3, 10 and 17 days), six fish from each group were taken from the respective boxes. These were subjected to physical and chemical analyses and sensory evaluation. The sensory and skin color evaluations were the first to be performed. The fish were then filleted and the pH, color, texture profile analysis (TPA), and lipid oxidation analyses were performed using the left-side fillet.

#### 2.3.1. pH, Color and Textural Parameters of Sea Bass Fillets

The pH values were measured using a portable instrument (Model HI 9025) with an electrode (FC 230C; both from Hanna Instruments, Villafranca Padovana, PD, Italy) and performing a two-point calibration (at pH equal to 7.01 and 4.01).

The colorimetric features (L* = lightness, a* = redness, b* = yellowness) of the fish fillets were determined using a Hunter Lab Miniscan™ XE Spectrophotometer (Model 4500/L, 45/0 LAV, 3.20 cm diameter aperture, 10° standard observer focusing at 25 mm, illuminant D65/10; Hunter Associates Laboratory Inc., Reston, VA, USA) by taking three readings for each sample along the left-side fillet (in correspondence of the cranial, middle and caudal fin regions). The instrument was normalized to a standard white tile provided with the instrument before performing analysis [22].

Rheological properties of the raw fish fillets were assessed using an Instron 5544 Universal Testing Machine (Instron Corp., Norwood, MA, USA). Texture Profile Analysis (TPA) was performed using a flat steel probe of 25 mm diameter, through a double compression test elaborated by the incorporated software. From the left side of each fish, three samples with a square surface (2 × 2 cm) and a height of 0.5 cm were excised in three different areas along the fillet (as described for color assessment). The mean values of measurements of each test per fish were retained for statistical analysis. The fillet samples underwent two compression cycles, 2 min apart, at a speed rate of 0.6 m/s. A compression equal to 75% of the initial height of the sample was performed for both cycles.

The following parameters were determined: hardness (N/cm^2^), expressed as the maximum force required to compress the sample; cohesion force resilience, that is the extent to which the sample could be deformed prior to rupture (A2/A1, where A1 and A2 are the total energy required for the first and second compression, respectively); springiness, (cm), i.e., the ability of the sample to recover its original form after the deforming force is removed; gumminess (N/cm^2^), that is the force needed to disintegrate a semisolid sample to a steady state of swallowing (hardness × cohesiveness); chewiness (N/cm), i.e., the work needed to chew a solid sample to a steady state of swallowing (springiness × gumminess) [23].

#### 2.3.2. Chemical, Fatty Acid Analysis and Lipid Oxidation of Sea Bass Fillets

Right-side fillets were rapidly chopped, combined in a pool, and homogenized for 1 min. AOAC procedures were used to assess the moisture, ether extract, raw protein, and the ash content [18]. The total lipids were extracted using a 2:1 chloroform/methanol (*v*/*v*) solution to determine the fatty acid profile [24]. The fatty acids were then methylated using a KOH/methanol 2N solution [25] and analyzed by gas chromatography (Shimadzu GC-17A) using a silicone–glass capillary column (70% Cyanopropyl Polysilphenylene-siloxane BPX 70 by Thermo Scientific, length = 60 m, internal diameter = 0.25 mm, film thickness = 0.25 µm). The starting temperature was 135 °C for 7 min, then increased by 4 °C/min up to 210 °C. Fatty acids were identified by comparison of retention times to authentic standards for percentage area normalization. Fatty acids were expressed as percentage (wt/wt) of total methylated fatty acids.

The food risk factors of meat were determined by calculating the Atherogenic (AI) and Thrombogenic (TI) Indices [26]:AI = [(C12:0 + 4 × C14:0 + C16:0)] ÷ [ΣMUFA + Σn-6 + Σn-3];
TI = [(C14:0 + C16:0 + C18:0)] ÷ [(0.5 × ΣMUFA + 0.5 × Σn-6 + 3 × Σn-3 + Σn-3)/Σn-6];
where MUFA are monounsaturated fatty acids.

Lipid oxidation was evaluated on fillet samples stored at 4 °C for 48 h after slaughtering by measuring the concentration of 2-thiobarbituric acid reactive substances (T-BARS) [27] and expressed as mg malondialdehyde (MDA)/kg meat.

#### 2.3.3. Sensorial Analysis Using QIM and Torry Scheme of Sea Bass

Sensory analysis was performed using the quality index method (QIM) and the Torry scheme, which are the most common procedures used for evaluating the freshness of raw and cooked fish, respectively [28]. Sensory analyses were carried out by eight panelists recruited from the Department of Soil, Plant and Food Sciences of the University of Bari “Aldo Moro”. These panelists were selected for their expertise in the descriptive analysis of food sensory parameters and for their experience with fish quality evaluation. Before the main evaluation, training sessions with sea bass fillets were conducted to train the panelists on how to use the QIM and Torry Schemes developed for the analysis of snakehead fish fillets [29].

The QIM method was based on a total of 12 demerit points describing five attributes among which were lightness of the skin, presence of hemorrhages, texture, eye characteristics as color and conformation, color and odor of the gills, color of the abdominal region and appearance of the anal region. Each panelist gave a score for each item and the sum of all the items was recorded. QIM ranged from 0 to 22 with lower scores reflecting premium quality of fish.

For the analysis of the odor and flavor of the cooked sea bass fillets, six slices (2 × 6 cm) were cut from the right-side fillets, wrapped in aluminum foil paper, placed in a perforated stainless-steel pan, and steam cooked for 10 min at 95–100 °C. After cooking, the samples were blind coded with a 3-digit random number and served to the panelists for evaluation using the Torry Scheme method developed by Shewan et al. [30], with some modifications made by Martinsdottir et al. [31] for medium fatty fish. An average score of ≤5.5 was used as the sensory rejection point, the Torry Scheme ranged from 3 to 10 with higher scores reflecting premium quality.

#### 2.3.4. Economic Efficiency

Benchmarking enables us to compare the efficiency of different production scenarios and to determine the best course of action. To explain the economic advantage following the use of olive oil and rosemary essential oil for aquaculture, it is proposed to estimate the efficiency of each diet, keeping all other production factors equal for all groups, which are assumed to be unchanged. Previous studies on sea bass farming used the benefit–cost ratio to analyze the economic performance of the different theses concerning feeding, intensity and farming system [32,33]. Efficiency, as the relationship between the result obtained and the resources employed, allows the actual performance to be measured. In a broader sense, efficiency (E) can be measured by the ratio of benefit (B) to cost (C).

Efficiency (E) assumes that the benefit obtained from the economic activity is greater than the cost (B > C) and the value of the product is greater than the value of the resources sacrificed in production [15]. The value of efficiency is significant when viewed in a comparative sense concerning the performances of the other theses. For this reason, the percentage variation rate of the efficiency of the experimental theses (FO + RO, OO, OO + RO) related to the efficiency of the baseline thesis (FO) was computed.

Below is the detail of the two parameters used in the efficiency analysis:For the comparison costs (C) computation, only the costs of feed with fish oil, olive oil, and rosemary essential oil used in the four experimental theses were taken into account, according to their current market prices and quantities used in the experimental theses;For the income (B) computation, values were derived from the sale of 1 kg of sea bass with a commercial size of 500 g;Additionally, in the case of the use of rosemary essential oil, the impact on revenue of the increased shelf-life was analyzed. A study on consumer preferences according to the expiry date of different food products [34] reported that consumers show a higher willingness to pay for perishable products that are stable for more days after their expiry date. Assuming an increase in the shelf-life of fish fed with rosemary essential oil, determined by the antioxidant action of the compounds detectable in fish, a minimum increase in revenue due to a better product positioning can be reasonably assumed. Therefore, an increase of 5% was applied for sea bass obtained with rosemary oil due to the extended shelf life.

### 2.4. Statistical Analysis

Data were analyzed using the SAS software 9.1 2004 (SAS Institute Inc., Cary, NC, USA) [35]. A 2 × 2 factorial design was used to examine the effects of the dietary oil (fish oil or olive oil), and the inclusion of rosemary essential oil.

The differences among groups were determined using Tukey’s test with *p*  <  0.05 as the significance level.

Flesh quality traits were analyzed by ANOVA for repeated measures with diet and rosemary essential oil inclusion as non-repeated factors, while storage time and their interaction as repeated factors. Results are reported as means and standard error of the means (SEM); significant effects were found at *p* < 0.05. When significant, means were compared using Student’s *t*-test.

## 3. Results

### 3.1. Growth Parameters

Table 3 shows the growth performances of European sea bass fed with the different diets. Sea bass fed with olive oil diets showed a significantly (*p* < 0.05) higher value of the Viscerosomatic (VSI) and Hepatosomatic indices (HIS) as compared to the fish fed with the diet containing only fish oil. As a consequence, the edible yield recorded in sea basses fed with olive oil was significantly lower in comparison with those containing FO (*p* < 0.05).

### 3.2. pH, Color, and Textural Parameters in Sea Bass Fillets

The results concerning pH and color of sea bass fillets are shown in Table 4. The pH values were affected only by the time of storage, showing a significant increase after 17 days of storage (*p* < 0.05). The diet did not affect the L value or the red index of fish skin, which were similar among groups. The inclusion of 200 ppm of RO in both feeds determined a significantly decrease (*p* < 0.05) in the yellow index (b*) values during 0, 3, and 17 days of analysis. Storage time affected the yellow index of fish skin: b* values increased significantly (*p* < 0.05) during storage in all groups.

The L* and a* indices of sea bass fillets were not affected by the diet and the storage time. The yellow index (b*) of the fish fillet was lowered (*p* < 0.05) by the inclusion of RO into the diet at days 0, 3 and 10 of storage. Within each diet, a significant increase (*p* < 0.05) of the b* index was found at day 0 as compared to days 10 and 17.

Table 5 shows the results concerning the textural parameters of sea bass fillets.

The addition of rosemary oil determined a significant (*p* < 0.05) increase in hardness at all sampling times. Furthermore, the hardness values significantly (*p* < 0.05) decreased in all the groups after 17 days of storage. The values of gumminess were influenced by the dietary oil (*p* < 0.05) and the time of storage (*p* < 0.05). At 17 days of storage, sea bass fed with olive oil diets showed a lower (*p* < 0.05) value of gumminess than the FO groups. Furthermore, the gumminess significantly (*p* < 0.05) decreased in all the groups after 10 and 17 days of storage.

The values of springiness and cohesion force resilience were unaffected by the type of dietary oil and the storage time.

The chewiness of fillets was significantly greater in the two groups fed with olive oil at all storage times (*p* < 0.05). Moreover, within each group, a significant decrease over time was observed (*p* < 0.05).

As for the MDA values, no significant differences among diets were observed during the first days of storage; the inclusion of rosemary oil determined a significant decrease in the concentration of MDA after 10 and 17 days of storage (*p* < 0.05). On the contrary, fillets from the FO and OO groups showed a significant increase in MDA concentration at days 0 and 3 as compared to day 17 (*p* < 0.05).

### 3.3. Chemical Composition and Fatty Acid Profile of Fillets

The chemical composition of fish fillets is shown in Table 6. No significant effect of the diet and time of storage was observed in any of the parameters analyzed.

Table 7 shows the fatty acid profile of the sea bass fillets. Fish fed with the diets containing fish oil showed fillets with a higher (*p* < 0.01) concentration of myristic and palmitic acids as compared to both the diets containing olive oil in all sampling times and the effects of the interactions O × S and RO × S were also significant *(p* < 0.05). These differences led to a significantly higher concentration of the total saturated fatty acids (SFAs), that was higher (*p* < 0.05) in the groups fed with fish oil. The time of storage influenced the concentration of myristic and palmitic acid in the fillets of sea basses fed without the inclusion of RO. After 17 days of storage, the concentration of these acids was significantly higher as compared to days 0 and 3. As for the total SFAs, the concentration at days 10 and 17 was significantly higher in comparison with that found at days 0 and 3 for the FO and OO groups. The inclusion of the essential oil only led to a significant increase in the concentration of SFAs at day 17 of storage, compared to day 0. 

At all sampling times, the concentration of oleic acid was significantly higher in the groups fed with olive oil diets (*p* < 0.01). Hence, the concentration of total MUFAs also followed the same trend (*p* < 0.01) and was lowered during storage (*p* < 0.05).

The concentration of linoleic acid was significantly greater in fillets from the groups fed with olive oil, regardless of the sampling time (*p* < 0.01).

The content of EPA and DHA acids in sea bass fillets was significantly higher in the groups fed with diets containing fish oil *(p* < 0.01). As a consequence, the amount of total PUFAs was also significantly greater (*p* < 0.01) in these groups and decreased markedly during storage *(p* < 0.05).

Within the PUFAs, a significantly higher concentration of n-6 fatty acids was found in fillets from the OO and OO + RO groups as compared to the other ones (*p* < 0.01). On the contrary, diets containing olive oil led to a significantly lower concentration of n-3 fatty acids, in all sampling times (*p* < 0.01). During storage, a significant decrease in the concentrations of total n-3 and n-6 fatty acids was observed *(p* < 0.05).

Finally, the n-3/n-6 ratio was significantly lower following feeding based on OO diets (*p* < 0.01).

The atherogenic and thrombogenic indices were unaffected by the diet and time of storage.

### 3.4. Sensory Analysis of Sea Bass

Table 8 shows the results of the sensory analysis of fish carried out by the QIM and TORRY methods. The inclusion of rosemary oil significantly lowered the QIM value only at day 0 (*p* < 0.01). Within each group, the QIM values significantly increased (*p* < 0.05) from day 0 to day 17.

The TORRY scores of cooked fillets from sea bass fed with the inclusion of RO in the diets were significantly higher (*p* < 0.05) at all sampling times. Within each group, a significant (*p* < 0.05) decrease in the index was recorded from day 0 to day 17.

### 3.5. Economic Efficiency of Sea Bass Feed

Table 9 shows the efficiency comparison of the four experimental diets taking into consideration the production of 1 kg of commercial size (500 g) sea bass. The formulation of the fish feed resulted in an estimated base feed cost of EUR 1.80 for one kg of the feed containing FO. Taking into consideration that 2 kg of feed are necessary to produce 1 kg of fish, the cost of feed needed to produce one kg of fish varies across the four experimental scenarios due to the different ingredient compositions, ranging from EUR 3.58 when using olive oil in partial replacement of fish oil up to EUR 3.70 when incorporating rosemary essential oil to fish oil.

The partial efficiency calculated on feed containing only fish oil (FO) is 2.78, while a slight improvement was observed following the use of olive oil (OO), with an increase of 0.56%. In contrast, the improvement in economic efficiency due to the addition of rosemary essential oil was 2.16 and 2.72%, respectively, for the FO + RO and the OO + RO groups. 

## 4. Discussion

The growth performances of the European sea bass were unaffected by the partial replacement of fish oil with olive oil. Accordingly, other researchers also stated that the partial replacement of fish oil with olive oil in the diet for European sea bass provided growth rates similar to those obtained when fish were fed on a 100% fish oil diet [37]. We found that the inclusion of olive oil in the sea bass diet led to greater Viscerosomatic (VSI) and Hepatosomatic (HSI) indices. Other authors [38] reported that a high content of dietary fat may increase the VSI and HIS indices as well as the content of lipids in the liver and visceral organs. Yilmaz et al. [39] found that the dietary inclusion of several essential oils, among which was rosemary, decreased the VSI and HSI indices of sea bass. In the present study, the lack of effect of rosemary oil on these indices can probably be attributed to the lower percentage of dietary inclusion as compared to that used by Yilmaz et al. (0.02 vs. 1%) [39]. Although no effect of the diet was found on the final body weight of sea basses, the edible yield was lower in fish fed the olive oil-based diets, as a consequence of a higher deposition of abdominal and liver fat.

The pH value of fillets is related to the post-mortem degradation processes and is widely influenced by the species, feeding regimen, season, etc. [40]. Volatile compounds are formed by the decomposition of nitrogenous compounds in the seafood, and this causes an increase in pH [41]. In the hake muscles, a pH increase was observed due to the accumulation of basic substances, such as ammonia and trimethylamine produced by microbial development [42]. In the present study, the addition of rosemary oil to the diets decreased the extent of pH variation; this may be due to its inhibitory effect on the growth of microorganisms, thus delaying the formation of volatile compounds [41]. The pH values found in our study are similar to those reported by other researchers in mirror carp and rainbow trout, respectively [43,44]. Özogul et al. [45] found a pH value increase during storage up to 7.38 at the sensory rejection (8th day of the storage); this value is similar to those found at 17 days of storage in our trial.

The present study analyzed the color changes that occurred in both the skin and flesh of sea bass. The color of skin is an important parameter which addresses consumers’ choice at purchase [46] and the presence of yellowish chromaticity is perceived as a lack of freshness. Pavlidis et al. [47] stated that the post-mortem changes in the color of fish skin are related to changes in light reflection as the result of cellular autolysis, denaturation of muscle proteins and/or changes in the intensity of pigments within the chromatophores. The pigmentation of the muscle in sea bass depends on the presence of proteins, carotenoids, and melanin [48]. In the flesh, the main factor involved in the change in chromaticity was associated with the b* parameter, indicating a tendency towards a more yellowish shade over time.

In this study, a noticeable influence of the storage time on hardness, gumminess, and chewiness was observed. The cohesion force, gumminess, springiness, and chewiness attributes decreased during the post-mortem cold preservation in all the experimental groups. Similar results were found by other authors [49,50] during the cold storage of sea bream. Post-mortem changes in fish texture are mainly caused by the modification of myofibrillar proteins, due to the protease action and to the variation of physical and chemical conditions [48].

Lipid oxidation is one of the important factors limiting the shelf-life of seafood products as well as the microbial degradation. Long-chain PUFAs, present in a high concentration in seafood lipids, are highly susceptible to oxidation, resulting in the formation of volatile compounds that have negative effects on the sensory quality of fish products and are commonly described as rancid odors and flavors [45]. Free radicals are known to induce lipid peroxidation, playing an important role in pathological processes in vitam as well as in the shelf-life of animal products. Thiobarbituric acid (TBA) is widely used for the assessment of the degree of lipid oxidation [51]. The consumability limit value of the TBA content was between 7 and 8 mg MDA/kg. In food suitable for consumption, the TBA values might reach the upper limit of 7 to 8 mg of MDA/kg [52]; in “perfect material”, the TBA value should be less than 3 mg of MDA/kg, and in “good material”, the TBA value should be no more than 5 mg of MDA/kg. The TBA values indicate the degree of rancidity of products, and values greater than 3–4 mg of MDA/kg indicate a loss of product quality [53]. In this study, the addition of rosemary oil proved to be effective in preserving fillets from oxidation from day 3 to day 17 of storage, providing MDA concentrations greatly below the limits reported for the perception of rancidity. Other studies also reported that the lowest TBA values were obtained following dietary treatment with essential oils, thus showing their antioxidant properties. A similar pattern of the increase in TBA has been reported in marinated sardine [54], pink shrimp [55], anchovy [56], and rainbow trout [57].

In our study, no significant influence of diet was found on the fillet chemical composition. Similar findings were reported by Ozogul in sea bass [45], by Seno-O in the Seriola quinqueradiata [58] and by Turchini in murray cod [59], who investigated the effects of substituting fish oil with olive oil. The lipid composition of sea bass flesh is quite variable since it depends on the dietary fat sources, the season, the water temperature, salinity, photoperiod, spawning, etc. [60,61]. Marine fish have a high dietary requirement for n-3 PUFAs; therefore, the diet of marine carnivorous fish contains high lipid levels, particularly n-3 highly unsaturated fatty acids (HUFAs) including eicosapentaenoic acid (20:5n-3, EPA) and docosahexaenoic acid (22:6n-3, DHA), because they are not capable of bio converting C18 PUFAs to C20 and C22 HUFAs which are essential and present in high concentrations in fish oil [62]. As documented in previous research, and in agreement with the present study, the fillet fatty acid profile was affected by the dietary oil source. Olive oil does not contain n-3 HUFAs, including EPA and DHA, while the main fatty acid is oleic acid. Our results are similar to those reported in fillets of murray cod fed with the total replacement of fish oil with olive oil [59]. Furthermore, similar evidence of the effects of the replacement of fish oil with olive oil were also reported in other fish species such as Seriola quinqueradiata [58] and Atlantic salmon [5]. In the present study, the best results in terms of reduction in the extent of lipid oxidation following the OO + RO diet may be explained by the synergic effect of rosemary oil and olive oil, which contains various antioxidants including oleuropein, hydroxytyrosol, tyrosol, α-tocopherol, and β-carotene [63].

The inclusion of rosemary oil lowered the QIM value only at day 0, when panelist scores were better for odor and appearance of fish supplemented with RO, regardless of the oil used in the diet. Similar results were described by Alvarez et al. [12] who investigated the quality index in sea bream fed with the supplementation of thyme essential oil and rosemary extract. At the end of the storage period, the demerit points were higher in the groups fed without RO, with values similar to those reported by Ozogul et al. [45] in a trial carried out on the sea bass. Limbo et al. [64] indicated that freshness of farmed sea bass was maintained for about 8 days for fish in melting ice (−0.5 °C); according to Chang et al. [65], the shelf-life of sea bass was 3 days when fish were stored at 5 °C and two weeks if they were stored at 0 °C. Cakli et al. [66] stated that the shelf-life of sea bass stored at 1.8 °C was 14 days. According to our results, sea bass fed with the inclusion of RO can be acceptably stored for up to 10 days, while at 17 days the fish are rejected.

The sensory analysis of cooked fish is necessary to determine consumer acceptability. The modified Torry freshness score was used to estimate the freshness of cooked fish, as a consequence of the evaluation of odor, taste, and texture. The TORRY scores of cooked fillets from sea bass fed with the inclusion of RO in the diets were higher at all the sampling times. Within each group, a noticeable decrease in the index was recorded from day 0 to day 17. Di Turi et al. [22], in a study on sea bass fed with or without the inclusion of RO, reported significant differences between the two diets at 10 days of storage; furthermore, the two curves (QIM and TORRY) defined one intersection point at day 14 of storage, after which the fillets are considered not acceptable.

The economic analysis of feed cost has shown the feasibility of using olive oil as a local feed ingredient in partial replacement of fish oil. As for the economic outcomes, it is important to assess the extent to which the impact of the enhanced benefit may be supported by the prolonged shelf-life. Just and Goddard suggested that consumer valuation is higher for applications that extend the duration of foods that have a more limited shelf-life, like the case of fish [67]. The analysis carried out in the present study suggests that the maximum minimum permissible rate of price escalation to offset the efficiency change is approximately 3%. This figure falls comfortably below the rate of price increase (5%) cautiously considered in the analysis, demonstrating that the base-line economic efficiency is assured even with a lower price increase.

## 5. Conclusions

The partial replacement of fish oil with olive oil in the sea bass diet has proven to be effective in terms of growth performance and fillet quality. Supplementation with rosemary essential oil delays lipid oxidation, thus improving the shelf-life of fish stored in ice by up to 10 days. Moreover, the addition of rosemary essential oil positively affects the sensory properties of fresh and cooked fillets. The economic analysis of feed cost has shown the feasibility of using olive oil as local feed ingredient in the partial replacement of fish oil and the potential willingness of consumers to pay for fish products endowed with prolonged shelf-life.

## Figures and Tables

**Table 1 animals-14-03237-t001:** Chemical constituents of *Rosmarinus officinalis* L. essential oil ^1^.

Compounds	Concentration
Hydrocarbons	40.00%
Limonene	2.705%
Linalool	1.728%
Eugenol	0.177%
Methyl Eugenol	0.070%
Geraniol	0.035%
Citronellol	0.021%
γ-terpinene	1.120%
Terpinolene	0.500%
Benzyl alcohol	0.001%
Lead	<3 mg/kg
Cadmium	<1 mg/kg
Mercury	<0.1 mg/kg
Arsenic	<1 mg/kg
Total eavy metals	<10 ppm

^1^ The essential rosemary oil was supplied by Farmalabor S.R.L. Italy.

**Table 2 animals-14-03237-t002:** Ingredients, chemical composition, and fatty acid profile of the diets.

Ingredients (%)	Dietary Treatment ^1^			
	FO	FO + RO	OO	OO + RO
Fish meal	60	60	60	60
Corn gluten	15	15	15	15
Wheat meal	8	8	8	8
Lysin (99%)	5	5	5	5
Fish oil	10	10	5	5
Rosemary oil	-	0.02	-	0.02
Olive oil	-	-	5	5
Premix ^2^	2	2	2	2
Chemical composition (% on DM basis)				
Moisture	10.04		10.95	
Crude protein	48.83		47.44	
Total lipid	18.42		18.87	
Ash	9.99		10.11	
Total Carbohydrates	12.72		12.63	
Gross Energy (MJ/kg)	19.16		19.31	
Fatty acid profile (%FA methyl esters)				
C14:00 (Myristic)	5.1		3.5	
C15:0 (Pentadecanoic)	0.5		0.5	
C16:0 (Palmitic)	15.8		13.4	
C17:0	5.1		2.1	
C18:0 (Stearic)	3.5		2.7	
C20:0	1.5		1.2	
C16:1 n-7 (Palmitoleic)	7.4		6.0	
C16:2 n-4	1.0		0.6	
C16:3 n-4	1.0		0.5	
C18:1 n-9 (Oleic)	13.5		37.3	
C18:1 n-7	4.3		4.1	
C20:1 n-9 (Eicosanoic)	3.2		2.5	
C18:2 n-6 (Linoleic)	5.3		10.4	
C20:2 n-6	0.6		0.9	
C18:3 n-6 (γ-linolenic)	1.1		0.1	
C18:3 n-3 (α-linolenic)	1.9		1.9	
C18:4 n-3 (Stearidonic)	1.6		0.9	
C20:4 n-3	0.7		0.4	
C20:5 n-3 (Eicosanoic, EPA)	10.9		6.9	
C22:5 n-3 (Docosapentaenoic, DPA)	0.9		0.8	
C22:6 n-3 (Docosahexaenoic, DHA)	10.2		8.3	

^1^ FO: fish oil; OO: olive oil; RO: rosemary oil; ^2^ Premix provides per kg: vitamin A (2,000,000 IU), Vitamin D3 (200,000 IU), Vitamin E (10,000 mg), Vitamin K3 (2500 mg), Vitamin B1 (3000 mg), Vitamin B2 (3000 mg), Calcium pantothenate (10,000 mg), Nicotinic acid (20,000 mg), Vitamin B6 (2000 mg), Vitamin B9 (1500 mg), Cupric sulphate (900 mg), Iron sulphate (600 mg), Potassium iodide (50 mg), Manganese oxide (960 mg), Sodium selenite (1 mg), Zinc sulphate (750 mg), Calcium carbonate (186,000 mg), Potassium chloride (24,100 mg), Sodium chloride (40,000 mg).

**Table 3 animals-14-03237-t003:** Growth performances of European sea bass fed diets containing fish oil or olive oil, with or without supplementation with rosemary oil.

Parameters	Dietary Treatment ^1^				SEM ^2^	*p*-Value ^3^		
	FO	FO + RO	OO	OO + RO		O	RO	O × RO
Final body weight (g)	580.00	584.14	586.00	588.92	8.73	0.136	0.074	0.121
Total body length (mm)	345.58	350.08	352.00	354.83	1.72	0.075	0.081	0.099
Standard length (mm)	304.50	307.95	311.00	310.08	1.54	0.081	0.096	0.093
Fork length (mm)	330.00	334.66	335.00	337.87	1.62	0.073	0.087	0.108
Condition factor (K)	1.97	1.99	1.97	1.96	0.02	0.084	0.091	0.116
Edible yield (%)	40.19	40.15	35.20	35.82	0.33	0.041	0.066	0.085
Viscerosomatic Index (%)	8.69	8.57	9.91	10.02	0.12	0.039	0.058	0.078
Hepatosomatic index (%)	1.69	1.85	2.20	2.12	0.04	0.044	0.087	0.089
Average daily weight gain (g/d)	1.30	1.32	1.33	1.35	0.01	0.083	0.073	0.125
Specific growth rate (%)	0.30	0.30	0.30	0.31	0.01	0.099	0.067	0.132
Survival (%)	94.70	95.12	94.78	94.35	0.08	0.087	0.091	0.117
Feed conversion rate (g/g)	1.98	1.99	1.98	1.97	0.02	0.069	0.089	0.136

^1^ FO: fish oil; OO: olive oil; RO: rosemary oil; ^2^ Standard Error of Means; ^3^ O: type of oil; RO: inclusion of rosemary oil; O × RO: interaction.

**Table 4 animals-14-03237-t004:** pH and color indices performed during ice storage on the dorsal skin area and on the fillets of sea bass fed the different experimental diets.

Parameters	Dietary Treatment ^1^																SEM ^2^	*p*-Value ^3^						
	FO				FO + RO				OO				OO + RO											
	0	3	10	17	0	3	10	17	0	3	10	17	0	3	10	17		O	RO	S	O × RO	O × S	RO × S	O × RO × S
pH	6.14	6.40	6.41	7.51	6.13	6.39	6.39	6.49	6.16	6.38	6.40	7.50	6.17	6.38	6.46	6.48	0.06	0.084	0.077	0.024	0.122	0.162	0.137	0.174
Color indices of Sea Bass dorsal skin area																								
L*	58.96	61.53	64.77	65.54	54.88	56.89	60.08	63.21	59.00	61.60	64.80	65.55	54.80	58.24	60.06	61.61	0.28	0.076	0.102	0.087	0.241	0.098	0.167	0.264
a*	−0.08	−0.15	−0.67	−0.79	−0.10	−0.80	−0.86	−1.07	−0.08	−0.15	−0.67	−0.80	−0.03	−0.79	−0.82	−1.37	0.19	0.089	0.208	0.101	0.631	0.142	0.134	0.146
b*	2.30	2.55	4.77	5.28	1.48	1.69	3.89	4.63	2.33	2.60 ^y^	4.77	5.29	1.77	1.92	3.67	4.42	0.33	0.061	0.022	0.031	0.098	0.105	0.044	0.133
Color indices of Sea bass fillets																								
L*	54.83	56.61	60.49	60.95	54.86	56.48	57.29	60.70	54.84	56.62	60.10	60.50	53.93	56.08	60.14	61.74	0.31	0.081	0.074	0.089	0.093	0.096	0.107	0.099
a*	−2.34	−3.10	−3.54	−3.69	−2.99	−3.03	−3.32	−4.15	−2.35	−3.12	−3.55	−3.69	−2.24	−3.12	−3.23	−3.58	0.16	0.074	0.082	0.096	0.099	0.087	0.102	0.114
b*	4.48	5.36	6.10	6.37	3.37	4.38	5.17	6.60	4.49	5.40	6.10	6.36	3.74	4.91	5.25	6.06	0.29	0.077	0.034	0.029	0.101	0.147	0.041	0.164

^1^ FO: fish oil; OO: olive oil; RO: rosemary oil; ^2^ Standard Error of Means; ^3^ O: type of oil; RO: inclusion of rosemary oil; S: storage; O × RO, O × S, RO × S, O × RO × S: interaction.

**Table 5 animals-14-03237-t005:** Textural parameters and MDA of fillets of sea bass fed the different experimental diets.

Parameters	Dietary Treatment ^1^																SEM ^2^	*p*-Value ^3^						
	FO				FO + RO				OO				OO + RO											
	0	3	10	17	0	3	10	17	0	3	10	17	0	3	10	17		O	RO	S	O × RO	O × S	RO × S	O × RO × S
Hardness (N)	22.43	21.30	17.04	14.19	27.31	26.66	23.94	21.64	23.14	20.89	19.56	15.74	28.07	26.43	23.45	21.12	3.84	0.112	0.028	0.034	0.117	0.124	0.123	0.118
Gumminess (N)	8.40	6.48	4.52	3.59	7.60	6.38	4.40	3.40	7.94	5.63	4.12	3.10	9.36	5.16	4.85	3.14	2.51	0.041	0.074	0.038	0.101	0.077	0.091	0.101
Springiness (mm)	2.79	2.93	3.25	2.38	2.83	3.03	3.50	2.30	2.78	2.94	3.27	2.83	3.06	3.30	3.29	2.80	0.22	0.065	0.071	0.068	0.097	0.101	0.144	0.128
Cohesion Force Resilience	0.56	0.41	0.25	0.13	0.53	0.28	0.25	0.11	0.54	0.26	0.25	0.12	0.55	0.51	0.34	0.23	0.01	0.061	0.063	0.077	0.104	0.112	0.137	0.109
Chewiness (N × mm)	23.71	16.45	11.02	10.02	23.66	16.32	10.72	10.13	28.45	18.73	13.00	11.60	28.90	18.60	14.03	11.89	4.38	0.039	0.087	0.047	0.099	0.101	0.094	0.148
MDA (mg/kg)	0.41	0.48	0.75	1.09	0.33	0.37	0.49	0.58	0.45	0.49	0.73	0.98	0.35	0.39	0.42	0.55	0.12	0.064	0.013	0.032	0.087	0.099	0.046	0.102

^1^ FO: fish oil; OO: olive oil; RO: rosemary oil; ^2^ Standard Error of Means; ^3^ O: type of oil; RO: inclusion of Rosemary oil; S: storage; O × RO, O × S, RO × S, O × RO × S: interaction.

**Table 6 animals-14-03237-t006:** Chemical composition of fillets of sea bass fed the different experimental diets.

Parameters	Dietary Treatment ^1^																SEM ^2^	Effects ^3^						
	FO				FO + RO				OO				OO + RO											
	0	3	10	17	0	3	10	17	0	3	10	17	0	3	10	17		O	RO	S	O × RO	O × S	RO × S	O × RO × S
Moisture	72.76	72.30	70.56	70.12	72.69	72.06	70.96	70.53	72.65	72.05	70.45	70.01	72.41	72.14	70.27	69.96	1.62	0.084	0.079	0.081	0.101	0.132	0.127	0.138
Crude Protein	19.07	19.20	20.01	21.02	19.02	19.00	20.05	20.84	19.40	19.21	20.69	21.10	19.38	19.32	20.20	21.13	0.69	0.064	0.059	0.067	0.079	0.076	0.088	0.101
Lipid	6.06	6.40	6.92	6.60	6.10	6.48	6.76	6.55	6.05	6.40	6.55	6.70	6.00	6.30	6.73	6.80	0.98	0.071	0.063	0.074	0.081	0.092	0.089	0.097
Ash	1.32	1.40	1.51	1.47	1.40	1.44	1.45	1.40	1.10	1.42	1.46	1.40	1.45	1.42	1.44	1.43	0.32	0.063	0.059	0.057	0.072	0.068	0.091	0.108
N-Free Extract	0.79	0.70	1.00	0.79	0.79	1.02	0.78	0.68	0.80	0.92	0.85	0.70	0.76	0.82	1.36	0.68	0.51	0.058	0.066	0.068	0.088	0.077	0.099	0.105

^1^ FO: fish oil; OO: olive oil; RO: rosemary oil; ^2^ Standard Error of Means; ^3^ O: type of oil; RO: inclusion of rosemary oil; S: storage; O × RO, O × S, RO × S, O × RO × S: interaction.

**Table 7 animals-14-03237-t007:** Fatty acid profile of fillets from European sea bass fed containing fish oil or olive oil with or without supplementation with rosemary oils (% Total Fatty Acids Methyl Esters) during storage.

Parameters	Dietary Treatment ^1^																SEM ^2^	Effects ^3^						
	FO				FO + RO				OO				OO + RO											
	0	3	10	17	0	3	10	17	0	3	10	17	0	3	10	17		O	RO	S	O × RO	O × S	RO × S	O × RO × S
C14:0 (Myristic)	5.86	5.88	6.21	6.86	5.86	5.88	5.95	6.36	4.95	4.98	5.60	6.13	4.93	4.94	4.96	5.33	0.12	0.006	0.057	0.024	0.074	0.038	0.047	0.088
C15:0 (Pentadecylic)	0.79	0.79	0.85	0.97	0.79	0.81	0.82	0.91	0.78	0.80	0.94	1.13	0.75	0.78	0.79	0.93	0.04	0.074	0.089	0.094	0.092	0.107	0.100	0.127
C16:0 (Palmitic)	20.86	21.55	22.46	23.02	20.86	21.09	21.36	22.02	18.50	18.51	19.06	19.55	18.53	18.56	18.91	19.68	0.51	0.005	0.038	0.031	0.081	0.041	0.046	0.092
C18:0 (Stearic)	4.67	4.68	4.94	5.09	4.67	4.68	4.71	4.99	4.55	4.56	5.25	5.56	4.54	4.56	4.61	5.01	0.11	0.067	0.063	0.059	0.074	0.077	0.069	0.072
Total SFA ^4^	32.18	32.90	34.46	35.94	32.18	32.46	32.84	34.28	28.78	28.85	30.85	32.37	28.75	28.84	29.27	30.95	0.66	0.003	0.041	0.017	0.062	0.039	0.040	0.084
C16:1 n-7	7.2	7.01	6.84	6.57	7.22	7.21	7.14	6.87	6.57	6.55	6.31	6.25	6.55	6.55	6.51	6.35	0.14	0.061	0.059	0.082	0.104	0.117	0.105	0.099
C16:1 n-9s	0.79	0.76	0.61	0.52	0.8	0.78	0.75	0.66	0.52	0.51	0.44	0.38	0.51	0.5	0.48	0.41	0.04	0.074	0.066	0.069	0.078	0.101	0.127	0.114
C18:1 n-7 (cis-vaccenic acid)	2.81	2.73	2.54	2.38	2.83	2.80	2.75	2.61	2.68	2.66	2.40	2.25	2.70	2.71	2.65	2.45	0.08	0.062	0.076	0.058	0.097	0.088	0.093	0.101
C18:l n-9 (Oleic)	20.01	19.98	19.73	19.56	20.00	19.96	19.93	19.65	28.84	28.82	28.46	28.40	28.85	28.82	28.78	28.51	0.87	0.002	0.062	0.066	0.091	0.089	0.102	0.116
C20:l n-9 (Eicosanoic)	3.98	3.97	3.84	3.66	3.95	3.93	3.92	3.85	4.59	4.55	4.38	4.3	4.58	4.58	4.55	4.43	0.09	0.071	0.067	0.064	0.079	0.091	0.094	0.097
Total MUFA ^5^	34.79	34.45	33.56	32.69	34.80	34.68	34.49	33.64	43.20	43.09	41.99	41.58	43.19	43.16	42.97	42.15	0.51	0.004	0.054	0.041	0.071	0.085	0.079	0.107
C18:2 n6 (linoleic)	4.12	4.10	3.98	3.84	4.13	4.11	4.08	4.00	6.93	7.13	6.91	6.58	6.90	6.89	6.87	6.75	0.22	0.007	0.062	0.058	0.074	0.089	0.093	0.099
C18:3 n-3 (α-linolenic)	0.94	0.91	0.89	0.81	0.90	0.90	0.90	0.87	0.82	0.80	0.77	0.70	0.80	0.80	0.80	0.75	0.04	0.062	0.058	0.059	0.084	0.086	0.094	0.118
C18:3 n-6 (γ-linolenic)	0.51	0.44	0.40	0.35	0.50	0.49	0.48	0.40	0.49	0.48	0.42	0.37	0.50	0.49	0.48	0.40	0.05	0.079	0.081	0.091	0.097	0.114	0.106	0.112
C18:4 n3	1.75	1.73	1.64	1.59	1.73	1.71	1.70	1.61	1.52	1.51	1.42	1.36	1.53	1.52	1.49	1.40	0.03	0.061	0.059	0.074	0.087	0.108	0.110	0.109
C20:4 n-3 (eicosatetraenoico)	0.64	0.61	0.52	0.48	0.65	0.64	0.60	0.54	0.37	0.36	0.30	0.27	0.40	0.38	0.37	0.31	0.06	0.084	0.067	0.088	0.091	0.097	0.125	0.162
C20:4 n-6 ARA	3.80	3.75	3.69	3.61	3.82	3.80	3.76	3.71	3.05	3.04	2.96	2.70	3.04	3.04	3.00	2.79	0.12	0.058	0.062	0.066	0.072	0.074	0.085	0.097
C20:5 n-3 (eicosapentaenoic, EPA)	8.98	8.93	8.74	8.67	8.92	8.90	8.88	8.79	6.69	6.65	6.56	6.45	6.70	6.68	6.65	6.57	0.31	0.003	0.066	0.057	0.069	0.071	0.083	0.099
C22:5 n-3 (docosapentaenoic, DPA)	1.40	1.37	1.35	1.31	1.43	1.40	1.40	1.37	1.10	1.09	0.95	0.87	1.12	1.11	1.10	0.95	0.08	0.057	0.064	0.066	0.078	0.081	0.086	0.104
C22:6 n-3 (docosahexaenoic, DHA)	10.89	10.81	10.77	10.71	10.94	10.91	10.87	10.79	7.05	7.00	6.87	6.75	7.07	7.09	7.00	6.98	0.72	0.004	0.072	0.071	0.081	0.088	0.093	0.117
Total PUFA ^6^	33.03	32.65	31.98	31.37	33.02	32.86	32.67	32.08	28.02	28.06	27.16	26.05	28.06	28.00	27.76	26.90	0.95	0.007	0.054	0.042	0.087	0.092	0.096	0.108
Total n-6 ^7^	8.43	8.29	8.07	7.80	8.45	8.40	8.32	8.11	10.47	10.65	10.29	9.65	10.44	10.42	10.35	9.94	0.98	0.004	0.053	0.037	0.076	0.068	0.091	0.101
Total n-3 ^8^	24.60	24.36	23.91	23.57	24.57	24.46	24.35	23.97	17.55	17.41	16.87	16.40	17.62	17.58	17.41	16.96	0.23	0.002	0.062	0.047	0.068	0.083	0.090	0.108
n-3/n-6	2.92	2.94	2.96	3.02	2.90	2.91	2.92	2.96	1.67	1.63	1.64	1.67	1.68	1.68	1.68	1.70	0.10	0.003	0.066	0.059	0.081	0.089	0.079	0.103
AI (Atherogenic Index)	0.71	0.71	0.64	0.67	0.62	0.64	0.60	0.66	0.57	0.57	0.66	0.71	0.67	0.66	0.56	0.58	0.02	0.058	0.061	0.067	0.085	0.091	0.097	0.114
TI (Thrombogenic Index)	0.31	0.34	0.34	0.35	0.31	0.32	0.32	0.34	0.35	0.36	0.40	0.43	0.36	0.36	0.37	0.40	0.01	0.061	0.066	0.074	0.088	0.079	0.091	0.118

^1^ FO: fish oil; OO: olive oil; RO: rosemary oil; ^2^ Standard Error of Means; ^3^ O: type of oil; RO: inclusion of rosemary oil; S: storage; O × RO, O × S, RO × S, O × RO × S: interaction; ^4^ Total SFA—saturated fatty acids (sum of C14:0 + C15:0 + C16:0 + C18:0); ^5^ Total MUFA—monounsaturated fatty acids (sum of C16:1 n-7 + C16:1 n-9 + C18:1 n-7 + C18:1 n-9 + C20:1 n-9); ^6^ Total PUFA—polyunsaturated fatty acids (sum of n-6 + n-3); ^7^ Total n-6 (sum of C18:2 n-6 + C18:3 n-6 + C20:4 n-6); ^8^ Total n-3 (sum of C18:3 n-3 + C18:4 n-3 + C20:4 n-3 + C20:5 n-3 + C22:5 n-3 + C22:6 n-3).

**Table 8 animals-14-03237-t008:** Sensory analysis of fillets of sea bass fed with different experimental diets.

Parameters	Dietary Treatment ^1^																SEM ^2^	Effects ^3^						
	FO				FO + RO				OO				OO + RO											
	0	3	10	17	0	3	10	17	0	3	10	17	0	3	10	17		O	RO	S	O × RO	O × S	RO × S	O × RO × S
QIM	4.08	4.75	9.13	12.11	3.23	5.25	8.97	11.97	4.10	4.77	9.16	12.14	3.25	5.36	8.93	12.28	4.18	0.061	0.027	0.036	0.087	0.088	0.052	0.097
TORRY	9.44	8.22	7.44	7.11	9.91	8.84	8.40	7.90	9.46	8.25	7.46	7.15	9.97	8.89	8.44	7.96	1.64	0.059	0.041	0.039	0.083	0.091	0.056	0.101

^1^ FO: fish oil; OO: olive oil; RO: rosemary oil; ^2^ SEM; ^3^ O: type of oil; RO: inclusion of rosemary oil; S: storage; O × RO, O × S, RO × S, O × RO × S: interaction.

**Table 9 animals-14-03237-t009:** Comparison of the efficiency of the four experimental diets.

Parameters ^2^	Dietary Treatment ^1^			
	FO	FO + RO	OO	OO + RO
Income (B) (EUR/kg fish ^#^)	10.00	10.50	10.00	10.50
Base feed cost (EUR/kgt)	1.80	1.80	1.52	1.52
OO cost * (EUR/kg feed)	--	--	0.27	0.27
RO cost (EUR/kg feed)	--	0.05	--	0.05
Total feed cost (EUR/kg)	1.80	1.85	1.79	1.84
Feed cost/kg fish (C) (EUR/kg)	3.60	3.70	3.58	3.68
Efficiency (E = B/C)	2.78	2.84	2.79	2.85
Efficiency variation rate (%)	--	2.16	0.56	2.72

^1^ FO: fish oil; OO: olive oil; RO: rosemary oil; ^2^ Source: our elaborations; ^#^ 1 kg of commercial size sea bass; * Virgin Olive Oil Price = 5.5 EUR/kg [36].

## Data Availability

Data are contained within the article.

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
