# Peer review of "Growth Performance and Flesh Quality of Sea Bass (Dicentrarchus labrax) Fed with Diets Containing Olive Oil in Partial Replacement of Fish Oil—With or Without Supplementation with Rosmarinus officinalis L. Essential Oil"

_animals, 2024, doi:10.3390/ani14223237_

Round 1

Reviewer 1 Report

Comments and Suggestions for Authors

The search for alternative raw materials to fish oil is relevant. However, olive oil has already been studied in this species. Starting from this point, the study has as a novelty the effect on the useful life that supplementation with rosemary can have.

L15

The relative importance of italy or a particular region of Italy in the olive oil production, has no relevance in this paper. The link between olive oil, and the mediterranean region and production of seabass could.

Lines 108-109 it says “Samples of all the diets were analysed for chemical composition [20] and for the fatty acid profile, and the results are shown in Table 2.”

The accuracy of the analyzes is excessive.  There is no variation between the analyzes with and without rosemary. This is normally due to the fact that the diets have not been analyzed separately, since it does not make sense since the inclusion of 0.02% of rosemary does not change the analyses. However, it is preferable that it be explained correctly. That is to say, one analysis was carried out for the diets based on FO and another for the diets based on OO instead of doing 4 analyses. Other option is that the result show in table 2 came from the raw materials analysis instead of the diet analysis.

Lines 122-123

It says “with temperature ranging from 17.5 to 18.5 °C, oxygen from 4 to 6 ppm” the temperature is suboptimal for growth and the oxygen may compromise also the growth.

Lines 130-131

It says “The growth trial lasted 200 days; at the end of the experiment, all fish were individually weighed, and 24 fish per diet were slaughtered. The fish were chosen from the three tanks corresponding to each experimental group (five subjects from each tank) and killed”

But 3 tanks *5 fish are 15 fish not 24  please clarify.

Line 142 It says : “Viscerosomatic Index (VSI) = {wet weight of viscera and associated fat / [wet body  weight − wet weight of viscera and associated fat]} × 100”

But VSI usually includes the viscera weight in denominator, why authors decide substract it?

Same issue in HSI formula

 Line 146  “Average weight gain (AWG) = [W2 - W1]/T,” since T is determining days,  it is Average daily weight gain to avoid confusions.

Lines 212-214  it says “Sensory analyses were carried out by eight panelists  recruited from the Department of Soil, Plant and Food Sciences of the University of Bari  “Aldo Moro”. These panelists were selected for their expertise in the descriptive analysis of food sensory parameters. ”

Usually to use just 8 panelist, it is need to be classified as expert panelist, with the specific training on that specie, I recommend clarify here that this was done, as authors explain  later in reference [31]

Line 233 and table 9

The procedure to elaborate the table 9 is not enough clear to me.

Tables

From my point of view p-value should not appear in the tables. Although it is common in many places to put it, its inclusion promotes the erroneous thinking that something is very close to being significant or is "VERY" significant. The authors should avoid discussion in function of p-value, and avoid references that could make think that that a lower p-value indicates higher differences.  If the p-value to determine differences is 0,05 maintain consistent through the paper.

Table 3

Please include initial body weight.

Feed conversion rate is (g/g) or unidimensional as you prefer but not (g/d)

Line 357.  Economic efficiency.

From my point of view this part is very important because finally, the inclusion  or not of a determined ingredient in the diet is strongly defined for its economic profitability. I encourage authors to improve this section with further information.

But I find the table 9 confusing,  Authors, claim a base cost of 360 €  but is not said for what?  Is that cost reflect the cost of ton? Of kilogram?

The OO cost  point to 10 kg of olive oils…  as it represent 5% of the diet  the cost should be for..  200 kg of diet… 

200 kg  of diet with a  FCR around 2 represent 100 kg of growth biomass of fish. With a benefit of 10.000 € , that is 100 € of profit by kilogram of fish.

For me there is something wrong or not explained

The exist parameter as ECR  (economic conversion rate) or EPI Economic profitability index, that would help to clarify.

All the information related to table 9 should be re-make and better explained.  The negative cost of the fish oil and extra cost of olive oil in the OO diet would indicate a price almost similar of the FO and OO

B is benefit? Profitability, ? understood as Income – production cost?  How it was estimated?

On the other hand the reduction of the % of edible yield part is  economically relevant.

Comments on the Quality of English Language

Taable 9  :  

Proceed (B) (€) 

You mean profitability??

Author Response

The search for alternative raw materials to fish oil is relevant. However, olive oil has already been studied in this species. Starting from this point, the study has as a novelty the effect on the useful life that supplementation with rosemary can have.

Dear Reviewer,

Thank you for your valuable comments and suggestions that lave led us to try to improve pur manuscript. We have argued our replies below.

L15

The relative importance of Italy or a particular region of Italy in the olive oil production, has no relevance in this paper. The link between olive oil, and the mediterranean region and production of seabass could.

We have reworded the sentence deleting the reference to Apulia and Italy.

Lines 108-109 it says “Samples of all the diets were analysed for chemical composition [20] and for the fatty acid profile, and the results are shown in Table 2.”

The accuracy of the analyzes is excessive. There is no variation between the analyzes with and without rosemary. This is normally due to the fact that the diets have not been analyzed separately, since it does not make sense since the inclusion of 0.02% of rosemary does not change the analyses. However, it is preferable that it be explained correctly. That is to say, one analysis was carried out for the diets based on FO and another for the diets based on OO instead of doing 4 analyses. Other option is that the result show in table 2 came from the raw materials analysis instead of the diet analysis.

Thank you, we have changed Table 2 by merging the two columns for the two diets.

Lines 122-123

It says “with temperature ranging from 17.5 to 18.5 °C, oxygen from 4 to 6 ppm” the temperature is suboptimal for growth and the oxygen may compromise also the growth.

Thank you for your comment, we meant to say that temperature never fell below 17.5-18.5 but the real range was 19 – 24, therefore, we have reworded the sentence. The oxygen level used seemed appropriate for growth according to previous experience and to other references.

Lines 130-131

It says “The growth trial lasted 200 days; at the end of the experiment, all fish were individually weighed, and 24 fish per diet were slaughtered. The fish were chosen from the three tanks corresponding to each experimental group (five subjects from each tank) and killed”. But 3 tanks *5 fish are 15 fish not 24 please clarify.

Thank you for your comment, the fish were 8 per tank and not 5, we have corrected the text.

Line 142 It says : “Viscerosomatic Index (VSI) = {wet weight of viscera and associated fat / [wet body  weight − wet weight of viscera and associated fat]} × 100” But VSI usually includes the viscera weight in denominator, why authors decide substract it? Same issue in HSI formula

The formulas have been corrected, we have carried out the measurements reported by Magnoni, L.J.; Gonçalves, O.; Cardoso, P.G.; Silva-Brito, F.; Ozório, R.O.; Palma, M., Viegas, I. Growth performance and quality indicators of European seabass (Dicentrarchus labrax) fed diets including refined glycerol. Aquaculture 2023, 569, 739377.

Line 146  “Average weight gain (AWG) = [W2 - W1]/T,” since T is determining days, it is Average daily weight gain to avoid confusions.

Thank you, we have corrected.

Lines 212-214 it says “Sensory analyses were carried out by eight panelists recruited from the Department of Soil, Plant and Food Sciences of the University of Bari “Aldo Moro”. These panelists were selected for their expertise in the descriptive analysis of food sensory parameters”.

Usually to use just 8 panelist, it is need to be classified as expert panelist, with the specific training on that specie, I recommend clarify here that this was done, as authors explain later in reference [31].

We have revised the text.

Line 233 and table 9

The procedure to elaborate the table 9 is not enough clear to me.

The table and the text in M&M and Results sections have been revised accordingly to your comments

Tables

From my point of view p-value should not appear in the tables. Although it is common in many places to put it, its inclusion promotes the erroneous thinking that something is very close to being significant or is "VERY" significant. The authors should avoid discussion in function of p-value, and avoid references that could make think that that a lower p-value indicates higher differences. If the p-value to determine differences is 0,05 maintain consistent through the paper.

We agree with you, however we have included the p-values in the table because other reviewers demand to report them. In this trial, we have considered the significance of the diet and of storage on the flesh quality traits, thus the inclusion of the p-values in the table might provide more detailed information.

Table 3

Please include initial body weight.

The initial body weight was 320 g for all the four groups and we have reported this data in Line 119, since we thought it was useless to repeat the same value four times.

Feed conversion rate is (g/g) or unidimensional as you prefer but not (g/d).

Thank you, corrected.

Line 357.  Economic efficiency.

From my point of view this part is very important because finally, the inclusion or not of a determined ingredient in the diet is strongly defined for its economic profitability. I encourage authors to improve this section with further information.

But I find the table 9 confusing, Authors, claim a base cost of 360 € but is not said for what? Is that cost reflect the cost of ton? Of kilogram?

The table and the text in M&M and Results sections have been revised accordingly to your comments

The OO cost point to 10 kg of olive oils…  as it represent 5% of the diet  the cost should be for. 200 kg of diet…

It is right. It corresponds to 200 kg of diet. We specified the amount in the table for more clarity

200 kg  of diet with a  FCR around 2 represent 100 kg of growth biomass of fish. With a benefit of 10.000 € , that is 100 € of profit by kilogram of fish.

For me there is something wrong or not explained.

We have corrected an evident error due to the shift of a value. The table and its commentary have been corrected. We also specify that this did not affect the discussion of the results as the analysis makes sense in relative terms.

The exist parameter as ECR (economic conversion rate) or EPI Economic profitability index, that would help to clarify.

to clarify.

We have added two literature references that use the cost-benefit ratio to estimate the breeding efficiency of sea bass.

  • Kumaran, M., Vasagam, K. K., Kailasam, M., Subburaj, R., Anand, P. R., Ravisankar, T., ... & Vijayan, K. K. (2021). Three-tier cage aquaculture of Asian Seabass (Lates calcarifer) fish in the coastal brackishwaters-A techno-economic appraisal. Aquaculture543, 737025.
  • Nhan, D. T., Tu, N. P. C., & Van Tu, N. (2022). Comparison of growth performance, survival rate and economic efficiency of Asian seabass (Lates calcarifer) intensively cultured in earthen ponds with high densities. Aquaculture554, 738151.

All the information related to table 9 should be re-make and better explained.  The negative cost of the fish oil and extra cost of olive oil in the OO diet would indicate a price almost similar of the FO and OO

The table and the text in M&M and Results sections have been revised accordingly to your comments

B is benefit? Profitability, ? understood as Income – production cost?  How it was estimated?

Income was estimated using the current selling prices of sea bass in the 500 g size range, considering a cautiously low price increase (5%) for those in which an extension of shelf life is assumed

On the other hand the reduction of the % of edible yield part is  economically relevant.

Table 9  : 

Proceed (B) (€)

You mean profitability??

We mean income.

Reviewer 2 Report

Comments and Suggestions for Authors This study aimed to investigate the effects of  olive oil and rosemary essential oil in partial replacement of fish oil on growth performances and flesh quality traits of farmed Dicentrarchus labrax. TThe design of this experiment was good, and there were abundant data and figures. However, some minor problem should be settled before accepted. 1. The economic efficiency part is better to be deleted, including the related method, results and discussion. Because the liturature cited (Ref. 17) is too old, and the results were only calculated by mathematics not related to any biological method. Also, this part was not so close to the theme of this study.   2. Some referecces were 20 years ago which were too old, better change into more recent ones. 

Author Response

This study aimed to investigate the effects of olive oil and rosemary essential oil in partial replacement of fish oil on growth performances and flesh quality traits of farmed Dicentrarchus labrax. The design of this experiment was good, and there were abundant data and figures. However, some minor problem should be settled before accepted. 1. The economic efficiency part is better to be deleted, including the related method, results and discussion. Because the literature cited (Ref. 17) is too old, and the results were only calculated by mathematics not related to any biological method. Also, this part was not so close to the theme of this study. 2. Some references were 20 years ago which were too old, better change into more recent ones.

Dear Reviewer,

Thank for your valuable comments; however, we would like to keep the economic description since in the introduction we described that one of the conditions for partial replacement of fish oil is also link to the economic profitability of the diet, due to the reduction of feed costs. Of course, the potential use of olive oil in partial replacement must be investigated under a biological and physiological point of view, in order to assess whether it may be used in the seabass diet without negative effects on growth performances or flesh quality.

We have carefully checked the references and deleted, when possible, old ones by replacing them with more recent citations. However, some bibliographical references to which you refer are obviously old as they firstly included efficiency in process analysis. In order to better contextualize the methodological approach, we included two literature references that use the cost-benefit ratio to estimate the breeding efficiency of sea bass.

  • Kumaran, M., Vasagam, K. K., Kailasam, M., Subburaj, R., Anand, P. R., Ravisankar, T., ... & Vijayan, K. K. (2021). Three-tier cage aquaculture of Asian Seabass (Lates calcarifer) fish in the coastal brackishwaters-A techno-economic appraisal. Aquaculture543, 737025.
  • Nhan, D. T., Tu, N. P. C., & Van Tu, N. (2022). Comparison of growth performance, survival rate and economic efficiency of Asian seabass (Lates calcarifer) intensively cultured in earthen ponds with high densities. Aquaculture554, 738151.

Reviewer 3 Report

Comments and Suggestions for Authors

1.    When writing any abbreviation for the first time, you should write it completely the first time. For example, RO and OO

2.    Introduction: The description in the preface is confused. The purpose of this paper is to investigate the effect of adding rosemary essential oil to the diet after adding fish meal on the growth and meat quality of fish. The preface has five paragraphs in total. The second and third paragraphs should be advanced, that is, the necessity of replacing fish meal with olive oil, the role of rosemary essential oil, and the importance of improving the meat quality of fish. Suggest merging the five paragraphs of the preface into three paragraphs.

3.    The references mentioned in the article are relatively old, it is recommended to use the latest five years of references

4.    L99 Please list the basis for adding olive oil and rosemary essential oil

5.    L100 Please list the storage conditions for the feed

6.    L125 Please write down how to feed and collect leftover materials to ensure the accuracy of FCR calculation and how to measure water quality

7.    L382 Reference 38 is not related to the proposed content

8.    L403-428 These two paragraphs of discussion only describe the changes of muscle indexes, not related to their own results, nor discuss the impact of adding rosemary essential oil on these indexes

9.    L449-451 This paragraph describes something very inexplicable, without specifying whether one's own results are similar or different from those studied by others.

10.  L465-466 The vegetable oil used in the 72nd reference does not include olive oil, please cite the reference reasonably

11.  L491 The economic analysis of feed cost is innovative

Author Response

Dear Reviewer,

Thank you for your valuable comments and suggestions that lave led us to try to improve our manuscript. We have argued our replies below.

  1. When writing any abbreviation for the first time, you should write it completely the first time. For example, RO and OO.

Thank you, we have reported the abbreviation in the very first lines of the abstract, highlighted in yellow.

  1. Introduction: The description in the preface is confused. The purpose of this paper is to investigate the effect of adding rosemary essential oil to the diet after adding fish meal on the growth and meat quality of fish. The preface has five paragraphs in total. The second and third paragraphs should be advanced, that is, the necessity of replacing fish meal with olive oil, the role of rosemary essential oil, and the importance of improving the meat quality of fish. Suggest merging the five paragraphs of the preface into three paragraphs.

We have deleted the first paragraph for greater conciseness. Now there are 4 paragraphs: the basis of fish oil replacement, the description of the choice of olive oil, the reason for rosemary essential oil supplementation and the explanation of the economic analysis, respectively.

  1. The references mentioned in the article are relatively old, it is recommended to use the latest five years of references.

We have carefully checked the references and deleted, when possible, old ones by replacing them with more recent citations.

  1. L99 Please list the basis for adding olive oil and rosemary essential oil

Thank you we have added a sentence in the introduction (lines 70-72) to better explain the reason of the association of rosemary essential oil

  1. L100 Please list the storage conditions for the feed

Done.

  1. L125 Please write down how to feed and collect leftover materials to ensure the accuracy of FCR calculation and how to measure water quality.

Thank you, we did not measure the amount of feed really ingested, but according to our experience sea bass were fed by hand until visual satiation to guarantee ad libitum feeding without the accumulation of leftovers that could have worsened water quality.

  1. L382 Reference 38 is not related to the proposed content

Thank you, we deleted the reference

  1. L403-428 These two paragraphs of discussion only describe the changes of muscle indexes, not related to their own results, nor discuss the impact of adding rosemary essential oil on these indexes

We have deleted some sentences that may seem speculative.

  1. L449-451 This paragraph describes something very inexplicable, without specifying whether one's own results are similar or different from those studied by others.

In these lines we say that “No significant influence of diet was found on the fillet chemical composition. Similar findings were reported by Ozogul in sea basses [45], by Seno-O in the Seriola quinqueradiata [67] and by Turchini in murray cod [68], who investigated the effects of substituting fish oil with olive oil”. Of course, we are describing our results in comparison with findings reported by other authors. For major clarity we have added “in our study”.

  1. L465-466 The vegetable oil used in the 72nd reference does not include olive oil, please cite the reference reasonably

We have deleted this reference.

  1. L491 The economic analysis of feed cost is innovative

Thank you for your comment, we are happy you have appreciated this kind of analysis.

Reviewer 4 Report

Comments and Suggestions for Authors

I have no real issues here with your manuscript in general. It is a worthwhile study and very topical. It is original work and provides an insight into the  use of this feed supplement for seabass. The focus on flesh quality is a good area with the consumer in mind. Linking this to diet and feeding is very useful and provides an interesting perspective to discuss. The study is well perfomed and with robust methodology and results. The introduction serves well its purposes and gives an overview of the general topic and literature base. The data is clear and with adequate statistical interpretation. The graphs, figures and tables give sufficient detail and are appropriate. Discussion is balanced and critical with a good assessment of the output and relevance to wider research aspects.  

Also please check this equation for hepatosomatic index 

Hepatosomatic index (HSI) = {wet weight of liver / [wet body weight - wet weight of liver]} × 100;

isn't it percent of liver weight to whole body weight? HSI (100 LW/BW) ? weigh fish complete first before removing liver and then remove liver (weigh liver) and use ratio  LW/BW*100

liver weight to whole fish weight ratio as percent

Comments on the Quality of English Language

The English is good and understandable. 

Author Response

I have no real issues here with your manuscript in general. It is a worthwhile study and very topical. It is original work and provides an insight into the  use of this feed supplement for seabass. The focus on flesh quality is a good area with the consumer in mind. Linking this to diet and feeding is very useful and provides an interesting perspective to discuss. The study is well perfomed and with robust methodology and results. The introduction serves well its purposes and gives an overview of the general topic and literature base. The data is clear and with adequate statistical interpretation. The graphs, figures and tables give sufficient detail and are appropriate. Discussion is balanced and critical with a good assessment of the output and relevance to wider research aspects.  

Also please check this equation for hepatosomatic index 

Hepatosomatic index (HSI) = {wet weight of liver / [wet body weight - wet weight of liver]} × 100;

isn't it percent of liver weight to whole body weight? HSI (100 LW/BW) ? weigh fish complete first before removing liver and then remove liver (weigh liver) and use ratio  LW/BW*100

liver weight to whole fish weight ratio as percent

Dear Reviewer,

Thank you for the appreciation of our research. As for the expression of the VSI index, we have corrected, thank you.

Round 2

Reviewer 1 Report

Comments and Suggestions for Authors

The viscerosomatic and hepatosomatic index have changed in material and methods but not the values ​​in table 3.

This produces a loss of credibility in the data.

Line 365 …

Avoid de use of quintal,  use kilograms instead.

I would recommend standardize measures,  income in table 9 should be in €/kg , based fish cost in €/kg OO Cost en RO cost would probably be mores understable un €/kg of diet…    so it would indicate how much of the diet cost correspond to olive oil  or to RO …

Part of the new text of point 3.5  should go to matherial and methods, and under the table 9. The table should be self explicative enough.

Author Response

The viscerosomatic and hepatosomatic index have changed in material and methods but not the values ​​in table 3. This produces a loss of credibility in the data.

Dear Reviewer, unfortunately it was a typo, and thanks to your comment we have corrected the mistake, that was a simple “copy and paste” error from another paper of ours. We guarantee you that the data are correct and that the indices have been calculated as reported. We kindly ask you to have no doubts about the correctness and reliability of the data reported.

Line 365 …

Avoid de use of quintal,  use kilograms instead.

We have corrected the text, it is highlighted in yellow, thank you.

I would recommend standardize measures,  income in table 9 should be in €/kg , based fish cost in €/kg OO Cost en RO cost would probably be mores understable un €/kg of diet…    so it would indicate how much of the diet cost correspond to olive oil  or to RO …

We have accepted your suggestion, the text has been reworded.

Part of the new text of point 3.5 should go to material and methods, and under the table 9. The table should be self explicative enough.

Done. We have also reorganized the structure of the table.

Reviewer 2 Report

Comments and Suggestions for Authors

The questons are well responded, and the reviesed version is more accepteable.

Also, one small question: I think the number of samples in each group should be displayed at the note of each table. 

Author Response

The questions are well responded, and the revised version is more acceptable.

Also, one small question: I think the number of samples in each group should be displayed at the note of each table.

Dear Reviewer, thank you for the appreciation of the revised version of our manuscript. We have accepted your suggestion and inserted the number of samples in each table, in relation to the sample analysed.

Reviewer 3 Report

Comments and Suggestions for Authors

The modification is very serious, and now there is only one small issue. The author of L97-106 still hasn't explained clearly the basis for adding RO/OO dosage. Why did they choose to add 0.02% RO and 5% OO?

Author Response

The modification is very serious, and now there is only one small issue. The author of L97-106 still hasn't explained clearly the basis for adding RO/OO dosage. Why did they choose to add 0.02% RO and 5% OO?

Dear Reviewer, thank you for the appreciation of the revised version of our manuscript.

As for the dosage of rosemary essential oil, in a preliminary experiment of ours carried out several years ago, we tested the dosage of 200 ppm because we observed that at this concentration the feed palatability was not affected by the presence of rosemary essential oil, that is very odorous. (Di Turi, L.; Ragni, M.; Caputi Jambrenghi, A.; Lastilla, M.; Vicenti, A.; Colonna, M.A.; Giannico, F.; Vonghia, G. Effect of dietary rosemary oil on growth performance and flesh quality of farmed seabass (Dicentrarchus labrax). Ital. J. Anim. Sci. 2009, 8, 857–859. https://doi.org/10.4081/ijas.2009.s2.857).

Another research carried out on seabass using oregano essential oil (Dinardo, F. R., Deflorio, M., Casalino, E., Crescenzo, G., & Centoducati, G. (2020). Effect of feed supplementation with Origanum vulgare L. essential oil on sea bass (Dicentrarchus labrax): A preliminary framework on metabolic status and growth performances. Aquaculture reports, 18, 100511) tested two dosages, i.e. 100 and 200 ppm, finding that that the dosage of 200 mg provided the best results in terms of fish growth.

With regard to the dosage of olive oil, we would like to highlight that the dosage used (5%) is the 50% of replacement of fish oil. Of course, higher percentages of replacement of fish oil may be investigated, but in this case, we have adopted a precautionary dosage.